# Neuromusculoskeletal modeling of spasticity: A scoping review

**Verônica Andrade da Silva**[1]*, **Rafael Lucio da Silva**[1], **Joseana Wendling Withers**[1],
**Kátia Janine Veiga Massenz**[1], **Maria Isabel Veras Orselli**[2], **Luciano Luporini Menegaldo**[3],
**Elisangela Ferretti Manffra**[1]

1 Health Technology Graduate Program, Pontifícia Universidade Católica do Paraná, Curitiba, Brazil,
2 Department of Biomedical Engineering, Faculdade Israelita de Ciências da Saúde Albert Einstein,
São Paulo, Brazil, 3 Biomedical Engineering Program, Universidade Federal do Rio de Janeiro, Rio de
Janeiro, Rio de Janeiro, Brazil

* veronica.andrade777@gmail.com

## Abstract

### Introduction

This scoping review aimed to provide an overview of neuromusculoskeletal models used to investigate the mechanisms underlying spasticity and identify issues to be addressed in future models.

### Materials and methods

We followed the Preferred Reporting Items for Systematic Reviews and Meta-Analysis Extension for Scoping Reviews (PRISMA-ScR) guidelines and searched four bibliographic databases (PubMed, Compendex Engineering Village, IEEE Xplore, and Science Direct). Inclusion criteria were original studies written in English that investigated the underlying mechanisms of spasticity in humans with no age restrictions. Two independent reviewers selected studies.

### Results

Eighteen studies met the inclusion criteria. Stroke was the neurological condition addressed by most studies, followed by cerebral palsy. The studies focused mainly on passive tasks with the knee joint as the primary target. All studies considered that spasticity was associated with alterations in the stretch reflex loop. Among the parameters tested by the studies, the reflex gains and thresholds were the parameters that could better represent levels of severity or effects of botulinum toxin type-A treatment. Recent studies proposed that stretching acceleration, muscle force, and force rate could be fed back into the feedback loop besides the muscle length and stretching velocity. However, no consensus was found among them. Finally, it has been that stiffness and viscosity of muscle-tendon-unit are also relevant for describing resistance to passive movement.

**Data availability statement:** All relevant data are within the paper and its Supporting Information files.

**Funding:** The authors acknowledge the scholarships from the National Council for Scientific and Technological Development (CNPq 400876/2019-1) and the research support. The authors also acknowledge the Pontifical Catholic University of Paraná. LLM also acknowledges support from FAPERJ and FINEP. The funders had no role in study design, data collection and analysis, decision to publish, or preparation of the manuscript.

**Competing interests:** The authors have declared that no competing interests exist.

## Conclusion

In order to provide relevant clinical and physiological information, future modeling should include supraspinal mechanisms in-depth, use image-based data to personalize non-neural parameters, specify models according to etiology and tasks, especially the active tasks of daily life activities.

## Introduction

Spasticity is a common consequence of upper motor neuron syndrome [1]. It occurs in several neurological conditions, such as stroke, cerebral palsy, multiple sclerosis, traumatic brain injuries, spinal cord injury, and others [2–4]. Spasticity has received considerable attention in literature due to its impacts on daily activities [5] and movement [3]. It might also lead to other comorbidities, such as joint contractures and deformities [5]. Despite its clinical relevance, the mechanisms underlying spasticity are not fully understood, and even its definition remains under debate [4,6,7].

The classical definition describes spasticity as "a motor disorder characterized by a velocity-dependent increase in muscle tone" [8], which is accompanied by "exaggerated tendon jerk, resulting from the hyperexcitability of the stretch reflex as one component of upper motor neuron syndrome" [8]. Although this definition is widely used, it has been criticized from several perspectives. First, it attributes hyperexcitable stretch reflexes to spasticity. However, other pathways, such as supraspinal control and even changes in motor neurons, may be involved in increased muscle activity during imposed stretches [4,9]. Another criticism is that this definition does not explicitly mention the involvement of sensory inputs [2,4,9,10].

More recently, the Support Program for Assembly of a Database for Spasticity Measurement (SPASM) defines spasticity as "disordered sensorimotor control resulting from an upper motor neuron lesion (UMNL) presenting as sustained or intermittent involuntary activation of the muscles" [4,11,12]. This definition covers the range of signals and symptoms collectively described as excessive features (positive signals), which include several forms of involuntary muscular hyperactivity, not only the hyperactive stretch reflex [13]. However, there is no consensus on the definition of spasticity [6,14], and its assessment is complex and controversial [15].

In clinical practice, spasticity is usually assessed with scales [15–18], such as the Ashworth Scale (AS) [19] and the Modified Ashworth Scale (MAS) [20]. AS or MAS is applied by a rater, passively stretching the target joint and assigning a score according to the perceived resistance to movement [20]. Therefore, it has been suggested that AS or MAS measure the joint hyper-resistance to passive movement rather than spasticity itself [21]. Joint hyper-resistance has both neural and non-neural components, with spasticity being one of the neural components [6,18,21]. The non-neural components are related to passive mechanical muscle-tendon tissue properties such as stiffness and viscosity [15,22,23]. The neural and non-neural components of joint hyper-resistance occur simultaneously [6,24]; identifying them individually is not trivial.



The neural mechanisms have a variety of origins at spinal and supraspinal levels [25]. Among spinal mechanisms, the hyperexcitability of alpha motoneurons stands out, amplified by persistent calcium ($Ca^{2+}$) and sodium ($Na^+$) currents, which generate prolonged depolarizations, known as plateau potentials [26]. In turn, supraspinal mechanisms are crucial in controlling spinal reflexes [1,25]. Lesions in premotor areas result in spasticity due to loss of inhibitory control over the bulbar reticular formation [1,25,27]. The medial reticulospinal tract, for example, facilitates spasticity and is crucial in maintaining spastic extensor tone. Together, these and other mechanisms form an intricate network of changes that promote spasticity.

A possible way to distinguish each component is by employing neuromusculoskeletal computational modeling, representing mathematically the musculoskeletal anatomy, the physiological mechanisms that produce muscle force, and the neural processes that command muscle contraction based on sensory information, intention, and a control law [28,29], to simulate human movement. The use of neuromusculoskeletal models enhances our understanding of the underlying mechanisms of neuromuscular function, complementing experimental methods. This approach not only provides estimates for variables that are difficult to measure but also enables 'what if' analyses, where cause-and-effect relationships can be inferred [28,30].

Considering that neuromusculoskeletal modeling has diverse possibilities and strategies, it is difficult for anyone wishing to embark on this path to build increasingly improved models without first having an overview of the topic. As far as we know, the most current reviews about spasticity focused on experimental quantitative methods that do not necessarily include neuromusculoskeletal computational modeling [6,18,31,32].

Thus, this scoping review aims to provide an overview of neuromusculoskeletal models used to investigate the mechanisms underlying spasticity and identify issues to be addressed in future models. To do so, we posed the following questions:

1. What physiological phenomena are represented in the neuromusculoskeletal models?

2. How do the models represent pathophysiology considering neural and non-neural contributions?

3. Which parameters have been able to quantify spasticity and distinguish severity levels?

## Methods

### Study design, protocol, and record

We followed the Preferred Reporting Items for Systematic Reviews and Meta-Analysis extension for Scoping Reviews (PRISMA-ScR) Checklist [33] (S1 Table) and registered our protocol on the Open Science Framework registration platform (https://osf.io/uqxpr/) under the https://doi.org/10.17605/OSF.IO/UQXPR.

### Search strategy

The search was performed in PubMed, Compendex Engineering Village, IEEE Xplore, and Science Direct. The following combination of keywords was defined as a search strategy: "muscle spasticity" AND "neuro-musculoskeletal model" OR "neuromusculoskeletal model" OR "neuro-musculoskeletal modelling" OR "neuromusculoskeletal modelling" OR "neuro-musculoskeletal modelling" OR "neuromusculoskeletal modeling" OR "neuromusculoskeletal simulation" OR "neuromusculoskeletal simulation." This search strategy was considered for the abstract, document title, and author keywords. The search strategy was similar in all databases, with adaptations to web interfaces (S2 Table). No date restriction was imposed. Searches in the databases took place until January 2024.

### Inclusion criteria

We included original studies written in English that used neuromusculoskeletal modeling to understand the underlying mechanisms of spasticity in humans. No restriction was imposed regarding the volunteers' age or the publication date.



## Exclusion criteria

The exclusion criteria were studies that used computational models to evaluate motor pathologies that could involve spasticity but did not focus on investigating the mechanisms of spasticity, review articles, letters to the editor, preprints, conference abstracts, books, and book chapters.

## Selection of studies

The article selection was organized in two stages. In the first stage, two reviewers (VAS and RFL) independently assessed the titles and abstracts of all references identified in the databases. Studies that did not meet the inclusion criteria were rejected. All references were managed using Microsoft Excel. At the end of the first stage, duplicate occurrences were excluded. The Kappa statistic was used to quantify the level of agreement during the selection process.

In the second stage, the same two reviewers read the remaining articles in full independently. The selected titles were compared, and a third author (JWW) was involved in case of disagreement. This third author read the article in full to decide on inclusion or exclusion; the final decision was made by consensus among the three authors.

To identify other studies that may have been missed during initial searches of electronic databases, we also chose to "identify studies through other methods," as recommended by the PRISMA [34]. To complement this, two reviewers (VAS and JWW) manually screened the reference lists of the selected articles and review articles related to modeling perspectives on spasticity. The articles found in this process were read in full independently by two reviewers (VAS and RLS) and included in this review, considering the same eligibility criteria and consensus procedure described earlier.

## Data extraction and analysis

After selecting the studies, four authors (VAS, RFL, JWW, KM) independently extracted relevant information, such as the physiological mechanisms addressed by the neuromusculoskeletal model, level of modeling (muscle or joint), motor task, variables fed back in the reflex loop, modeled muscles, parameters that quantified neural and non-neural components of joint hyper-resistance and, if existing, their differences according to the severity of spasticity or healthy condition.

We also extracted information not directly related to modeling, namely, authorship, year of publication, etiology of spasticity, the severity of spasticity and clinical assessment employed, sample sizes, and age characteristics of spasticity and healthy groups, if existing, the target joints according to groups investigated, and experimental data collected. The results are presented through summary tables.

To assess the potential of each model to reproduce experimental data, we adapted the physiological plausibility score proposed by Davico et al. [35] to our context. The original score assigns one point per model and participant whenever one of the following criteria is met: (i) good tracking of the joint moment with Coefficient of Determination ($R^2$) values ≥ 0.7, (ii) good tracking of muscle excitations ($R^2 \geq 0.5$), (iii) maximum knee joint contact forces not exceeding 3.5 body weight, and (iv) inclusion of accurate musculoskeletal anatomy (from medical images) [35]. We excluded criterion (iii) because we did not consider contact forces. We observed that the selected studies used different indicators for the goodness of fit besides $R^2$. Thus, we added the Variance Accounted For (VAF) and Cross-Correlation Coefficients (CCC) in criteria (i) and (ii), with values above 80% [36] and 0.7, respectively [37]. Moreover, we have added two new criteria to characterize the different contributors of spasticity and the tissue changes that may occur secondary to spasticity.

Thus, the adapted physiological plausibility score has the following criteria:

(i)   good tracking of experimentally observed joint moments with $R^2 \geq 0.7$ or CCC $\geq 0.7$ or VAF $\geq 80\%$.

(ii)  good tracking of experimentally observed muscle activity (EMG signals) with $R^2 \geq 0.5$, or CCC $\geq 0.7$, or VAF $\geq 80\%$.

(iii) inclusion of personalized musculoskeletal anatomy of the evaluated subjects with medical images.



(iv) distinguishes the neural origin of spasticity: spindle hypersensitivity (due to increased firing rate of muscle spindles) from motor neuron hyperexcitability.

(v) distinguishes neural from non-neural components of joint hyper-resistance.

For criteria (i) and (ii), points were attributed according to reported mean values of $R^2$, CCC or VAF among participants. Several revised studies evaluated the models for multiple conditions (e.g., task, velocity, group) and more than one muscle. Thus, we assigned one point per criterion for each condition whenever the criterion was met. The total points ($TPmodel$) of the model were given by Equation 1:

$$TPmodel = Points_i + Points_{ii} + Points_{iii} + Points_{iv} + Points_v \qquad (1)$$

where $Points_i$, corresponds to the number of points achieved in criterion (i), and so on. Studies that determined, for instance, the goodness of fit in several conditions could receive one point for each condition when criteria (i) or (ii) were met. To avoid bias, favoring models tested in several conditions, we defined a normalization factor $(NF)$, which is the number of conditions tested. The sum of the points ($TPmodel$) of the model was then divided by the normalization factor $(NF)$, resulting in a normalized physiological plausibility score ($Pscore$), given by Equation 2,

$$Pscore = TP / NF \qquad (2)$$

This score provides normalized metrics that allow physiological plausibility to be compared among models, regardless of the number of conditions tested.

## Results

A total of 522 articles were identified in the databases, and 12 additional articles were retrieved through other methods, as shown in the PRISMA flow diagram (Fig 1). After removing duplicates, 302 articles from the databases were assessed for eligibility based on title and abstract, resulting in twenty-eight full-text articles being reviewed. Of these, eight studies met the inclusion criteria [38–45]. Additionally, all twelve articles retrieved through other methods were reviewed in full, with ten meeting the eligibility criteria [46–55]. Thus, eighteen articles were included in this review. The strength of agreement between reviewers was almost perfect (Kappa k = 0.959).

The distribution regarding country, experimental data collected, target joints and characteristics of study groups is shown in Fig 2. Experimental data were collected in sixteen studies, and all reported kinematics [38–43,45–52,54,55]. Muscle activation signals were collected in most studies [38–43,45,47–52,54], while force or torque [40,42,51,52,55], ground reaction forces [39,43,45,50], and medical images [43] appeared less frequently.

### Representation of physiological components

The block diagram of a generic neuromusculoskeletal system to investigate the spasticity of a target muscle or muscle group is provided in Fig 3. The supraspinal structures, i.e., motor cortex, brainstem, cerebellum, and basal ganglia communicate with the spinal cord via the descending and ascending tracts. The spinal cord hosts the pools of alpha (α) and gamma (γ) motor neurons (MN) whose activity depends on supraspinal signals and proprioceptive feedback provided by primary (Ia), secondary (II) afferents from muscle spindles, and Ib afferents from the Golgi tendon organ (GTO). The muscle-tendon unit (MTU) is described as a typical Hill-type muscle model composed of the contractile element (CE) and passive elements (PE), illustrating tissue stiffness and viscosity. The force produced by muscle contraction is transferred to the bones as a tendon force, generating joint torques and movement.

The CE comprises extrafusal muscle fibers innervated by α-motor neurons [56]. The muscle spindles (intrafusal fibers) run parallel to the extrafusal fibers. They are innervated by γ-motor neurons [57,58] whose action prevents slackening



**Fig 1. PRISMA ScR flow diagram: Scope review process for neuromusculoskeletal models of spasticity.**

during muscle shortenings and enables continuous regulation of spindle sensitivity [58,59]. The central regions of the intrafusal fibers are in contact with Ia and II afferents [56,58], which are stretch sensory receptors. The Ia afferents encode muscle length and velocity of muscle stretching, and the II afferents encode only muscle length [59,60]. As the Ia afferents are more sensitive to velocity than length, it is said that they provide a sense of movement, while II afferents give a sense of position [59,60].

The GTOs are encapsulated mechanoreceptors found at the musculotendon junctions in series with the extrafusal fibers [61–63]. They are innervated by Ib afferents and communicate with inhibitory interneurons in the spinal cord [59,62]. According to the literature, the GTO carries information about the muscle force and muscle force rate, creating a negative feedback system [25,59,61,62]. The purpose of this system is to regulate muscle stress by decreasing muscle activation when significant forces are produced [60,63] and, therefore, to safeguard the integrity of muscles.

None of the reviewed papers represented the whole neuromusculoskeletal system, and Table 1 shows the physiological components described in each model.

The MTU of a target muscle or muscle group (lumped actuator) is present in all reviewed papers and is represented by a Hill-type muscle model with CE and PE, which convert muscle activations into forces or torques. In turn, activation dynamics, converting neural excitation into muscle activation, was often represented as a first-order differential equation [40,42,44,52,55].

                                                  

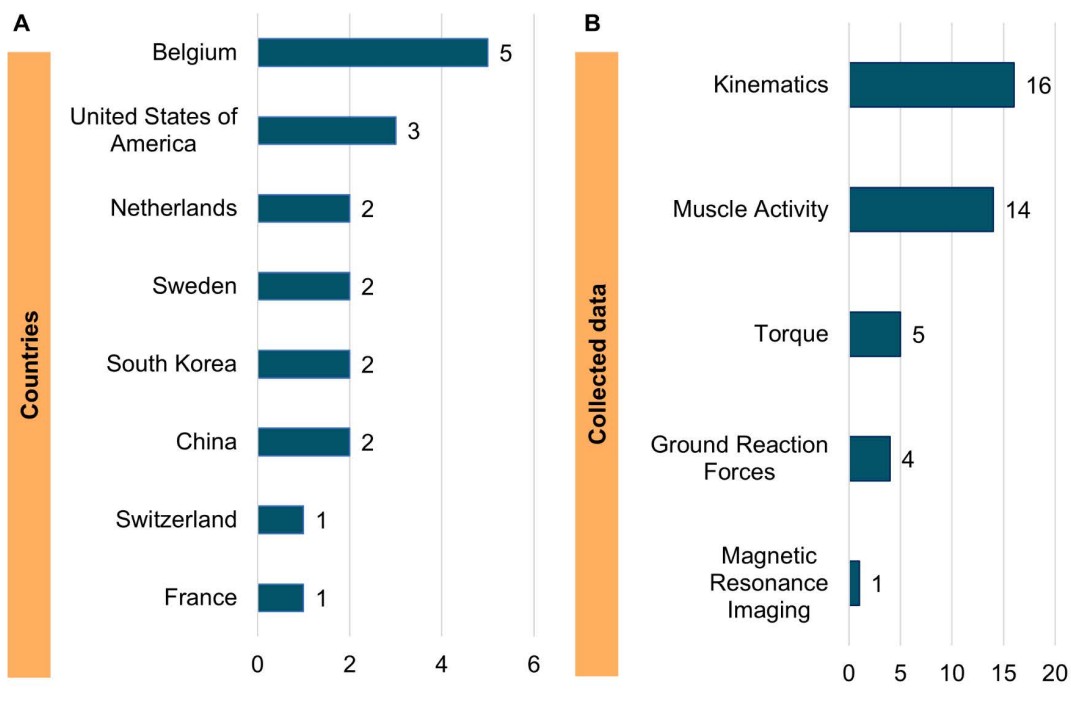

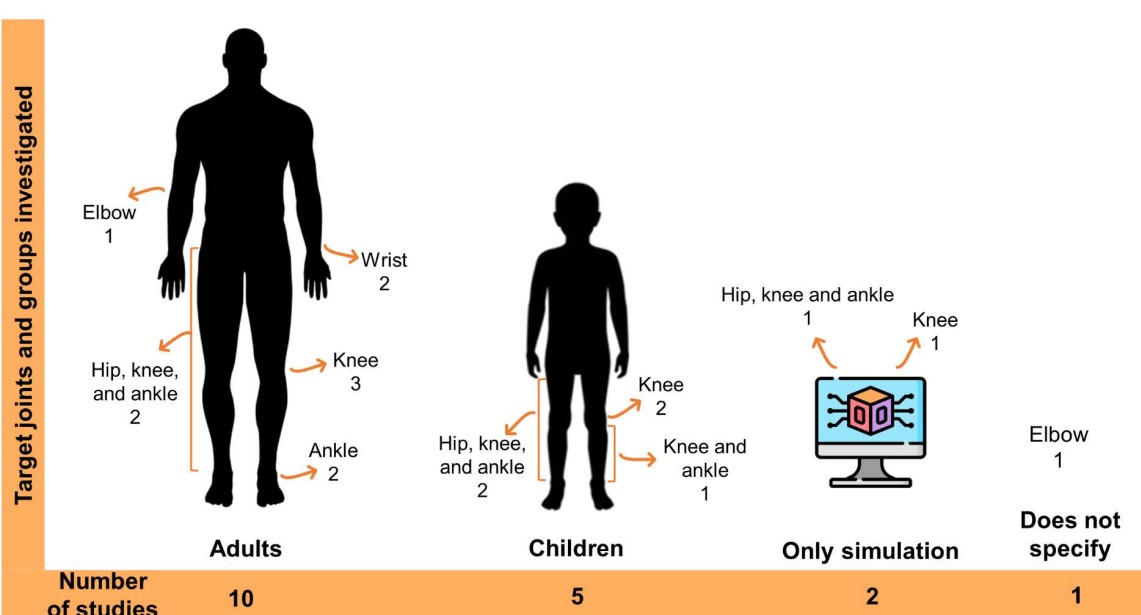

**Fig 2. Distribution of studies according to the countries, experimental data collected, target joints, and sample characteristics.**

Most models represented Ia and II sensory afferents encoding velocity and length [38–41,44–47,50,52], following the clinical definition of spasticity as a velocity-dependent increase in the tonic stretch reflex [1,8,15].

One representation of sensory afferents was the $v^{0.6}$ hybrid model [40,42,52,55], originally based on the firing characteristics of the cat muscle spindle during normal locomotion [64,65]. The spindle firing rate is a function of muscle fiber length (or joint angle), muscle stretching velocity (or angular velocity), and a constant background discharge rate [40,42,52,55]. The

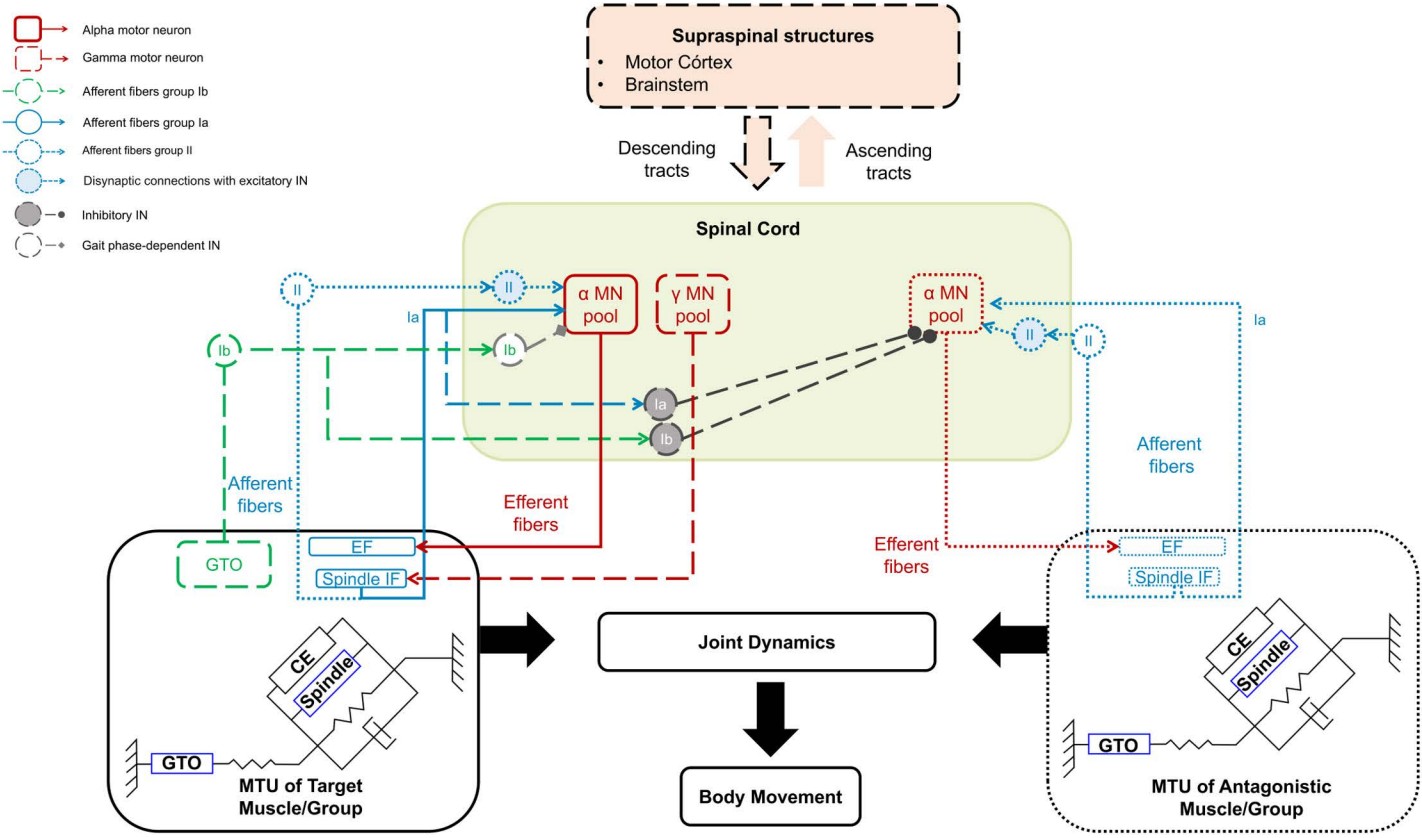

**Fig 3. Block diagram of components represented in neuromusculoskeletal models of spasticity.** Solid lines indicate components represented by all models, dotted lines denote elements represented by some studies, and long dashed lines indicate representation in a single study. At the top, supraspinal structures are represented in a simplified form, usually as a neural drive. Ascending tracts were not represented in the reviewed models. The spinal cord is represented by the pool of α motoneurons (MN) and γ motoneurons (MN) from the target muscle/group and its antagonist and interneurons from groups Ia, II, and Ib. Proprioceptive feedback is provided by primary (Ia), secondary (II) afferents from muscle spindles, and Ib from the Golgi tendon organ (GTO). Most neuromusculoskeletal models represent only the muscle-tendon unit (MTU) of the target muscle/group, i.e., where the spasticity is manifested. A few models also include the MTU of antagonistic muscles. The MTUs are typically represented by a Hill-type model, which consists of the contractile element (CE), or extrafusal fibers (EF), representing the contractile properties of the muscle fibers, parallel passive elements with viscoelastic properties, and intrafusal fibers (IF). These elements are arranged in series with the tendon, forming the pennation angle, which refers to the angle between the muscle fibers and the tendon.

position and velocity feedback have different gains, namely static and dynamic gains. The stretch reflex loop has a delay, representing the interval between the onset of mechanical stretch and the onset of the corresponding firing. The Gaussian cumulative distribution function proposed by Fuglevand et al. [66] models the α-motoneuron pool [40,42,52,55] as an input-output relationship. The input is the spindle firing rate, and the output is the muscle activation. Although γ-motoneuron is not explicitly modeled, the authors considered that the static and dynamic gains of the $v^{0.6}$ hybrid model would represent its activation level [40,42,52,55]. It is important to mention that all synaptic inputs (except those from the spindle) to alpha and gamma motoneurons in the identified models, when present, are assumed to be constant [40,42,52,55].

Another strategy for representing sensory receptors was the implementation of reflex controllers in OpenSim. Firstly, a very simplified controller that depended on the feedback variables of the reflex loop [39,41] and, more recently, a reflex circuitry adapted from Geyer & Herr's model [44,45].

In other studies, the feedback was represented by first-order dynamics to model the time delay and characterized by a threshold and a gain factor [43,50]. It is essential to highlight that some studies have proposed spasticity models



Table 1. Physiological components represented in the studies.

| Authors | Target MTU | Antago- nistic MTU | α MN pool | γ MN pool | Afferent Ia feedback | Afferent II feedback | Afferent Ib (facilitation) | Afferent Ib (recriprocal inhibiton) | Viscoelasticity elements in the MTU | Supraspi- nal Struc- tures |
|---|---|---|---|---|---|---|---|---|---|---|
| He; Norling; Wang, 1997 [46] | ● | ● | ● | | ● | ● | | | | |
| He, 1998 [38] | ● | ● | ● | | ● | ● | | | ● | |
| Feng; Mak, 1998 [47] | ● | ● | ● | | ● | ● | | | ● | |
| Le Cavorzin et al., 2001 [54] | ● | | ● | | ● | | | | ● | |
| Fee; Foulds, 2004 [48] | ● | ● | ● | | ● | | | | ● | |
| Koo; Mak, 2006 [40] | ● | | ● | ● | ● | | | | | |
| De Vlugt et al., 2011 [51] | ● | ● | ● | | | | | | ● | |
| Kim; Eom; Hase, 2011 [49] | ● | | ● | | ● | | | | ● | |
| Jansen et al., 2014 [39] | ● | | ● | | ● | ● | | | | |
| van der Krogt et al., 2016 [41] | ● | ● | ● | | ● | | | | ● | |
| Wang et al., 2017 [42] | ● | | ● | ● | ● | | | | ● | |
| Wang; Gäverth; Herman, 2018 [55] | ● | | ● | ● | ● | | | | ● | |
| Falisse et al., 2018 [50] | ● | ● | ● | | ● | | | | | |
| de Groote et al., 2018 [53] | ● | | ● | | ● | | | | ● | |
| Shin et al., 2019 [52] | ● | | ● | | ● | ● | | | ● | |
| Falisse et al., 2020 [43] | ● | | ● | | ● | | | | ● | |
| Bruel et al., 2022 [44] | ● | ● | ● | | ● | ● | ● | ● | ● | |
| Veerkamp et al., 2023 [45] | ● | ● | ● | | NS | NS | | | | ● |

*Note.* The black dots denote the physiological components represented in each study. MTU stands for muscle-tendon-unit. NS indicates that the study did not specify the type of feedback.

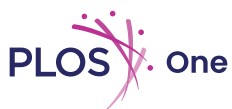

suggesting that instead of length and stretching velocity, the relevant feedback variables would be force (or torque) and its first derivative [50,53]. These studies compared models with different feedback variables: length, velocity, force, and the first derivative of force, aiming to gain insights into the physiology behind spasticity.

The GTO is explicitly modeled only in the work of Bruel et al. [44] for the gait test. It provided excitatory feedback to the α-motor neurons of the target muscles during the stance stage of gait (direct Ib facilitation to the extensors) and inhibitory feedback to the antagonist's muscles, modeling reciprocal inhibition [44]

Several studies have investigated the non-neural contributions that may co-occur with spasticity by associating them with parameters in the Hill-type MTU models [38,41–44,49,51,52,54,55].

The only work that explicitly models the supraspinal drive is Veerkamp et al. [45], which adds a constant term to the force, position, and velocity feedback terms.

## Characteristics of the neuromusculoskeletal models

Table 2 presents the characteristics of neuromusculoskeletal models based on the level of modeling (muscle or joint) and variables in the reflex loop. By joint level, we mean that the model considered the net joint torque without identifying muscles' individual contributions. Most reviewed studies represented muscles individually [38–41,43–47,50,52]. Regarding the variables fed back in the reflex loop, muscle stretching velocity/joint velocity is present in most of the models (n = 16) [38–42,44–50,52–55], followed by the muscle fiber length/joint position (n = 13) [38–40,42,44–47,50,52–55]. A few recent models considered muscle force/joint torque [43–45,50,53] and force/torque rate [43–45,50,53] feedback variables. Only one model fed back muscle stretching acceleration [43–45,50,53].

In Table 2, we also present the parameters related to the neural (spasticity-related) and non-neural contributions to joint hyper-resistance and how they were adjusted to represent healthy and pathological conditions. Parameters of the reflex loop were included in all the reviewed studies as associated with the absence or presence of spasticity and its severity.

In studies that employed the $v^{0.6}$ hybrid model [40,42,52,55], MN pool parameters were associated with the intensity of MN response to spindle firing. These parameters are the MN pool threshold, corresponding to the minimum firing rate to generate neural excitation, and the MN pool gain, related to the passive membrane properties and the voltage-sensitive membrane conductance of motoneurons [40,42,52,55]. The motoneuron threshold [40,42,52,55] is a lumped parameter that reflects the net excitatory and inhibitory inputs to the motoneuron pool [40,42,52,55].

Two studies included neural parameters not related to the reflex loop, namely, a baseline torque and a short-range stiffness torque constant [53] and a representation of the supraspinal drive [45].

The role and range of non-neural parameters were addressed in 10 out of the 18 reviewed articles [38,41–44,49,51,52,54,55]. Most of these studied parameters are related to passive elements of the MTU, such as damping coefficients or linear and nonlinear stiffness. Unlike this mainstream, Kim, Eom, and Hase [49] also considered segmental inertial parameters. More recently, two studies evaluated the role of the muscle force-length relationship parameters optimal muscle fiber length [43,44] and muscle fiber maximal isometric force [44]. In the study of Falisse et al. [43], tendon slack length and optimal pennation angle were also addressed.

Regarding the methods employed by the authors to vary parameter values, detect sensitive variables and calibrate the models, most of the studies employed optimization techniques [40–45,48–52,55]. Until 2001, varying parameters manually was the most usual approach [38,46,47,54], but recent studies with complex models have also explored parameters in this manner [39,44,53].

A mapping of muscle modeling across the primary references is shown in Fig 4. Most studies have used Hill-type muscle models to represent the MTU [38–40,43,46,47,50,52] or other formulations based on Hill-type muscle models [41,44,45]. In the most recent studies [39,41,43,44,50,53], the authors employed Hill-type muscle models using OpenSim software [67]. Some authors combined several individual muscles into lumped muscular actuators [42,48,49,51,53,55]. In other studies, there was a combination of individual muscles and lumped muscular actuators [38,39,41,44–47,52]. In He,



**Table 2. Characteristics of neuromusculoskeletal models and parameters associated with spasticity and other contributors of joint hyper-resistance.**

| Authors | Level of Modeling | Variables fed back in the reflex loop | Reflex neural parameters | Non-reflex neural parameters | Non-neural parameters |
|---|---|---|---|---|---|
| He; Norling; Wang, (1997) [46] | Muscle | Muscle fiber length Muscle stretching velocity | Muscle length threshold (M) Stretching velocity threshold (M) Length and velocity feedback gain (M) | None | None |
| He (1998) [38] | Muscle | Muscle fiber length Muscle stretching velocity | Muscle length threshold (M) Stretching velocity threshold (M) Length and velocity feedback gain (M) | None | Damping coefficient (M) Linear stiffness coefficient (M) |
| Feng; Mak (1998) [47] | Muscle | Muscle fiber length Muscle stretching velocity | **Muscle length threshold (M) Stretching velocity threshold (M) Length and velocity feedback gain (M)** | None | None |
| Le Cavorzin et al. (2001) [54] | Joint | Joint angular position Joint angular velocity | **Position-dependent velocity threshold (M)** Time delay in reflex loop (electromechanical coupling delay and reflex loop latency) (M) | None | **Damping coefficient (M)** |
| Fee; Foulds (2004) [48] | Joint | Joint angular velocity | **Velocity feedback gain (O)** Time delay in reflex loop (O) **Reflex torque onset time (O)** | None | None |
| Koo; Mak (2006) [40] | Muscle | Muscle fiber length Muscle stretching velocity | MN pool parameters (O) | None | None |
| De Vlugt et al. (2010) [51] | Joint | None | EMG weighting factors (O) Activation cut-off frequency (O) | None | **Damping coefficient (O) Linear stiffness coefficient (O)** |
| Kim; Eom; Hase (2011) [49] | Joint | Joint angular velocity | **Joint velocity feedback threshold (O) Velocity feedback gain (O)** | None | Damping coefficient (O) Nonlinear stiffness coefficients (O) |
| Jansen et al. (2014) [39] | Muscle | Muscle fiber length Muscle stretching velocity | **Length feedback gain (M) Velocity feedback gain (M)** | None | None |
| van der Krogt et al. (2016) [41] | Muscle | Muscle stretching velocity | Velocity feedback gain (O) | None | **Nonlinear stiffness coefficient (O) Passive muscle strain due to max isometric force (O)** |
| Wang et al. (2017) [42] | Joint | Joint angular position Joint angular velocity | Position feedback gain (O) Velocity feedback gain (O) **MN pool parameters (O)** | None | **Damping coefficient (O) Linear stiffness coefficient (O) Nonlinear stiffness coefficient (O)** |
| Wang; Gäverth; Herman (2018) [55] | Joint | Joint angular position Joint angular velocity | Position feedback gain (O) Velocity feedback gain (O) **MN pool parameters (O)** | None | Damping coefficient (O) Linear stiffness coefficient (O) **Nonlinear stiffness coefficient (O)** |

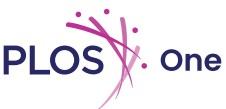

**Table 2.** (Continued)

| Authors | Level of Modeling | Variables fed back in the reflex loop | Reflex neural parameters | Non-reflex neural parameters | Non-neural parameters |
|---|---|---|---|---|---|
| Falisse et al. (2018) [50] | Muscle | MO1: Muscle fiber length and muscle stretching velocity<br>MO2: Muscle force and muscle force rate (first derivative of muscle force)<br>MO3: Muscle fiber length, muscle stretching velocity, and muscle stretching acceleration | Length feedback gain (O)<br>Velocity feedback gain (O)<br>Force feedback gain (O)<br>Force rate feedback gain (O) | None | None |
| De Groote et al. (2018) [53] | Joint | MO1: Joint angular position and joint angular velocity<br>MO2: Joint torque and torque rate (first derivative of joint torque) | Position feedback gain (M)<br>Velocity feedback gain (M)<br>Torque feedback gain (M)<br>Torque rate feedback gain(M) | Baseline torque (M)<br>Short-range stiffness (SRS) torque constant (M) | None |
| Shin et al. (2020) [52] | Muscle | Muscle fiber length<br>Muscle stretching velocity | **MN pool parameters (O)** | None | **Damping coefficient (O)<br>Nonlinear stiffness coefficients (O)<br>Shape parameter of passive torque (O)** |
| Falisse et al. (2020) [43] | Muscle | Muscle force<br>Muscle force rate (first time-derivative of muscle force) | Force feedback gain (O)<br>Force rate feedback gain (O) | None | Optimal muscle fiber length (O)<br>Tendon slack length (O)<br>Optimal pennation angle (O)<br>Maximal fiber contraction velocity (O) |
| Bruel et al. (2022) [44] | Muscle | Muscle fiber length<br>Muscle stretching velocity<br>Muscle force | Length feedback gain and threshold (O)<br>Velocity feedback gain and threshold (O)<br>Force feedback gain and threshold (O) | None | Muscle fiber maximal isometric force (M)<br>Optimal muscle fiber length (M) |
| Veerkamp et al. (2023) [45] | Muscle | MO1: Muscle fiber length and muscle stretching velocity<br>MO2: Muscle force | Velocity feedback gain (O)<br>Force feedback gain (O) | Supraespinal drive (O) | None |

*Note.* Methods for varying parameters: (M) manual adjustment; (O) optimization. MO1, MO2 and MO3 indicate that the same study developed three different models. Parameters that could distinguish between healthy and pathological conditions or among levels of severity are highlighted in bold. MN parameters are threshold, corresponding to the minimum firing rate to generate neural excitation, and the MN pool gain, related to the passive membrane properties and the voltage-sensitive membrane conductance of motoneurons.

Norling, and Wang [46], for example, the biceps femoris, semitendinosus, and semimembranosus muscles were modeled as lumped muscular actuators, so the three muscles have the same color in Fig 4 (orange).

## Motor tasks and experimental conditions

In Table 3, we present the motor tasks, the etiology of spasticity, and the characteristics of the experimental groups of volunteers. Spasticity was predominantly investigated during passive tasks [38,40–42,46–49,51–55], and the only voluntary task investigated was gait [39,43–45,50]. The passive tasks were the pendulum test focusing on the knee [38,46,48,49,53,54] or elbow joint [47], passive stretching at a controlled velocity [40,42,52,55] and passive stretching with a controlled duration [41,51]. Two studies included gait and passive stretching [43,50] in their investigation. The etiology





| Muscles | He; Norling; Wang, 1997 | He, 1998 | Jansen et al., 2014 | van der Krogt et al., 2016 | Falisse et al., 2018 | Shin et al., 2020 | Falisse et al., 2020 | Bruel et al., 2022 | Veerkamp et al., 2023 | Feng; Mak, 1998 | Koo; Mak, 2006 |
|---|---|---|---|---|---|---|---|---|---|---|---|
| **Type of muscle model** | Hill | Hill | Hill | Other formulations | Hill | Hill | Hill | Other formulations | Other formulations | Hill | Hill |
| Gluteus maximus | | | | | | | | ● | ● | | |
| Iliopsoas | | | | | | | | ● | ● | | |
| Rectus femoris | ● | ● | ● | ● | ● | | | | ● | | |
| Vastus medialis | ● | ● | ● | ● | ● | | | ● | ● | | |
| Vastus lateralis | ● | ● | ● | ● | ● | | | ● | ● | | |
| Vastus intermedius | | | ● | ● | ● | | | ● | ● | | |
| Biceps femoris long head | ● | ● | | ● | ● | | ● | ● | ● | | |
| Semimembranosus | ● | ● | | ● | ● | | ● | ● | ● | | |
| Semitendinosus | ● | ● | | ● | | | ● | ● | ● | | |
| Biceps femoris short head | ● | ● | | ● | | | | ● | ● | | |
| Tibialis anterior | | | | | | | | ● | ● | | |
| Gastrocnemius medialis | ● | ● | ● | ● | ● | ● | ● | ● | ● | | |
| Gastrocnemius lateralis | ● | ● | ● | ● | ● | ● | ● | ● | ● | | |
| Soleus | | | ● | | | ● | | ● | ● | | |
| Biceps brachii long head | | | | | | | | | | ● | ● |
| Biceps brachii short head | | | | | | | | | | ● | ● |
| Brachialis | | | | | | | | | | ● | ● |
| Brachioradialis | | | | | | | | | | ● | ● |
| Long head of triceps brachii | | | | | | | | | | ● | |
| Lateral head of triceps brachii | | | | | | | | | | ● | |
| Medial head of triceps brachii | | | | | | | | | | ● | |

**Fig 4. Mapping of the modeled muscles.** The presence of a rectangle in a muscle row indicates that it was a target muscle in the model(s) developed by the authors in the corresponding column. Rectangles of the same color indicated that muscles were represented in a lumped way.

of spasticity varied widely across studies, and most included only adults in the experimental sample (see Fig 2C). Two studies comprised numerical simulations without experiments [44,53].

The pendulum test was initially proposed to evaluate knee extensors' spasticity [68,69]. However, one reviewed study adapted the test to the elbow joint [47]. This well-established clinical test relies on kinematic data to assess spasticity quantitatively using gravity to elicit stretch reflexes during passive swinging of the limb [68,70]. Volunteers are instructed to relax completely while the evaluator fully extends the knee or the elbow and then suddenly releases it so the limb can swing freely [68,71]. The resistance to passive stretch changes the kinematics of pendular movement [68,71]. Performing the test with appropriate instrumentation, such as the use of surface EMG, combined with neuromusculoskeletal modeling allows the assessment of the contributions of neural and non-neural components, such as muscle viscoelastic properties that dampen oscillatory movement [46–49,53]. The distinction between neural and non-neural contributions can be achieved by incorporating parameters representing various physiological mechanisms [46–49,53]. Advanced parameter estimation techniques [48,49] or carefully calibrated manual adjustments [47,53,54] can be used to fit the model to experimental data. This approach enables the quantification of both neural (e.g., reflex activity) and non-neural parameters, providing insights into their relative contributions to joint hyper-resistance.

Passive stretching at constant velocity was another task during which neural and non-neural parameters were determined by combining experimental data and modeling. Two studies used the NeuroFlexor [72] (Aggro MedTech AB, Solna, Sweden), which imposed wrist extension movements at two constant velocities (slow and fast) [42,55]. Simultaneously,

**Table 3. Motor tasks, etiology of spasticity, and characteristics of experimental groups (when present).**

| Studies | Tasks | Etiology | Severity of spasticity | Sample Size Spasticity Group | Sample Size Healthy Group | Age (Mean ± SD) Spasticity Group | Age (Mean ± SD) Healthy group |
|---|---|---|---|---|---|---|---|
| He, Norling and Wang (1997) [46] | Pendulum test of the knee joint in supine and sitting postures | Multiple sclerosis | AS: 0–4 | 46 | 0 | 25–55 Years (Range) | – |
| He (1998) [38] | Pendulum test of the knee joint in supine and sitting postures | Multiple sclerosis | AS: 0–4 | 59 | 0 | 26–68 Years (Range) | – |
| Feng and Mak (1998) [47] | Pendulum test of the elbow joint | Pure spastic quadriplegia | Not evaluated | 3 | 1 | Not described | Not described |
| Le Cavorzin et al. (2001) [54] | Pendulum test of the knee joint at sitting posture | Stroke (7), incomplete spinal cord injury (6), syringomyelia (1), myelitis (1) | MAS: 1,53 (Mean) | 15 | 8 | 34,9 ± 14,0 Years | 34,9 ± 14,0 Years |
| Fee and Foulds (2004) [48] | Pendulum test of the knee joint in the sitting position with a backrest reclined at 15° from the vertical. | Cerebral Palsy | No spasticity and mild spasticity | 2 | 1 | 9 Years | 9 Years |
| Koo and Mak (2006) [40] | Passive stretching of the elbow flexors at a constant angular velocity (80∘ s−1) | Not described | MAS: +1 | 1 | 0 | 44 Years | – |
| De Vlugt et al. (2010) [51] | Ramp-and-hold ankle dorsiflexion of 0.25, 0.5, 1, and 2 seconds (s) duration | Stroke | AS: 0–3 0,94 ± 1,07 (Mean ± SD) | 19 | 7 | 63,6 ± 8,5 Years | 55,4 ± 10,3 Years |
| Kim, Eom and Hase (2011) [49] | Pendulum test in supine position with release angles of 90°, 70°, and 50° by vertical line | Stroke | MAS: 1–1+ | 10 | 0 | 54,0 ± 10,1 Years | – |
| Jansen et al. (2014) [39] | Walking at 1 km/h | Stroke | Not applicable | 0 | 1 | – | 21 Years |
| van der Krogt et al. (2016) [41] | Passive stretching of the hamstrings at slow (> 5 s) and fast speed (< 1 s) | Cerebral Palsy | MAS: 1–2 IPSA: left leg hamstrings | 11 | 9 | 11,5 ± 3,4 Years | 11,0 ± 3,2 Years |
| Wang et al. (2017) [42] | Passive stretching of the wrist flexors at slow (5° s−1) and fast (236° s−1) velocities | Stroke | MAS: 0–4 2,76 ± 1,20 (Mean ± SD) | 17 | 17 | 50,0 ± 11,0 Years | 48,0 ± 10,0 Years |
| Wang, Gäverth and Herman (2018) [55] | Passive stretching of the wrist flexors at slow (5° s−1) and fast (236° s−1) velocities | Stroke | MAS: 0–4 2,61 ± 1,20 (Mean ± SD) | 21 | 0 | 50,0 ± 12,0 Years | – |
| Falisse et al. (2018) [50] | Gait Passive stretching of the hamstrings and gastrocnemii at slow (10° and 15°/s), medium (75° and 55°/s), and fast (200° and 100°/s, respectively) velocities | Cerebral Palsy | MAS: 1–2 IPSA: medial hamstrings and gastrocnemii | 6 | 0 | 11,7 ± 2,9 Years | – |
| De Groote et al. (2018) [53] | Pendulum test of the knee joint (only simulation) | Cerebral Palsy | Not applicable | 0 | 0 | Not applicable | Not applicable |

*(Continued)*

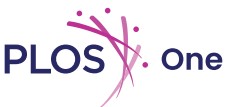

| Studies | Tasks | Etiology | Severity of spasticity | Sample Size Spasticity Group | Sample Size Healthy Group | Age (Mean ± SD) Spasticity Group | Age (Mean ± SD) Healthy group |
|---|---|---|---|---|---|---|---|
| Shin et al. (2020) [52] | Passive stretching of plantar flexors at slow (near 15°/s) and fast velocities (near 150°/s) | Stroke (6), Ankle injury (1), Peroneus weakness (1), Foot drop (2) | Not evaluated | 10 | 10 | 60,2 ± 17,9 Years | 26,9 ± 1,2 Years |
| Falisse et al. (2020) [43] | Walking at a self-selected speed. Passive stretching of hamstrings and gastrocnemii of the right side throughout the full range of motion (at slow and fast velocities) | Cerebral Palsy | MAS: 0–2 (according to muscle) IPSA: right medial hamstrings and gastrocnemii | 1 | 0 | 10–15 Years (Range) | – |
| Bruel et al. (2022) [44] | Gait (only simulation) | Not described | Not applicable | 0 | 0 | Not applicable | Not applicable |
| Veerkamp et al. (2023) [45] | Walking at a self-selected speed | Cerebral Palsy and Hereditary Spastic Paraplegia | SPAT: 0–5 | 17 | 0 | 8,2 ± 3,34 Years | – |

*Note.* SD is the Standard Deviation. AS refers to the Ashworth Score. MAS is the Modified Ashworth Score. IPSA indicates Instrumented Passive Spasticity Assessment which consists of a clinical assessment of spasticity performed by an examiner who imposes passive stretches. SPAT is a Spasticity Test.

this device measures the resistance force. The contribution of the elasticity and viscosity components could be estimated by measuring resistance during slow and fast velocities [42,55]. The neural contribution to spasticity corresponds to the difference in strength between the resistance to fast movement minus the resistance to slow movement, assuming that in the slow movement, the neural contribution was negligible [42,55,72]. Passive stretching at controlled velocity was also performed using an isokinetic dynamometer [40] and a customized device for measuring torque [52].

Another way to perform passive stretching was by fixing the range of motion and the duration of movement either by rotating the limb manually [41] or with the help of an electrical device [51]. In both cases, slow and fast movements were performed [51].

In most of the reviewed studies [38,40,42,43,46,49–51,54,55], the severity of spasticity was determined subjectively by clinical measures, such as AS and MAS. Still, some authors also used the Instrumented Passive Spasticity Assessment (IPSA) [41,43,50] and Spasticity Test (SPAT) [45].

The AS has four levels of severity: where (0) indicates no increase in tone, (1) a slight increase in tone presented by a "catch" when the limb is moved in flexion/extension, (2) a more marked increase in tone, but is easy to move the limb through the full range of motion, (3) indicates a considerable increase in tone with challenging to move the limb passively, and (4) designates a limb rigid presentation in flexion and extension [19]. The MAS levels 0, 2, 3, and 4 follow the same description as AS [20]. MAS level 1 denotes a slight increase in tone visible by a catch, release, or minimal resistance at the end of the range of motion when the limb is moved in flexion or extension [20]. In MAS, there is an additional level 1+, corresponding to a slight increase in tone presented by a catch and minimal resistance throughout less than half of the range of motion [20].

The IPSA consists of a clinical assessment of spasticity performed by an examiner who imposes passive stretches [73]. During the examination, measurement instruments such as surface electromyography (sEMG), inertial measurement units

to track joint angles, and force sensors to verify the forces applied by the examiner are used [73]. Finally, SPAT has five levels where (0) denotes no spasticity, (1) indicates an increase in muscle resistance somewhere in the range of motion, (2) indicates an increase in muscle resistance with a clear catch and a release, (3) indicates an increase in muscle resistance with a clear catch without release, (4) indicates clonus with less than 5 beats, and (5) denotes clonus with 5 beats or more [74].

## Model parameters and levels of spasticity and other contributions of joint hyper-resistance

In this topic, we present how the model parameters have been associated with the absence or presence of spasticity and other contributions of joint hyper-resistance and its severity level in the experimental groups.

Regarding reflex neural parameters, length and velocity feedback thresholds were higher in the healthy group than in the group with spasticity in the work of Feng and Mak [47], while the feedback gain was smaller. Similarly, Le Carvozin et al. [54] found a greater position-dependent velocity threshold in healthy individuals than those with spasticity. In contrast, Fee and Foulds [48] found that velocity feedback gain was smaller in healthy conditions in comparison to spastic conditions. More recently, Jansen et al. [39] also found that length and velocity feedback gains were smaller in the healthy group than in the hemiparetic gait simulation. In the work of Kim, Eom, and Hase [49], a higher gain was observed in the group with more severe spasticity than in the group with a milder one.

Differences in the MN pool parameters among severity levels [42] and treatment conditions [55] were also observed. MN pool gain was larger, and its threshold was smaller, the more severe the spasticity [42]. MN pool threshold increased after 4 weeks of botulinum toxin Type-A (BoNT-A) application and returned to baseline levels after 12 weeks [55]. In the work of Shin et al. [52], the authors divided the volunteers into two groups: those with reflex torque higher than passive torque and those with the opposite. The first group exhibited a lower MN pool threshold than the second [52]. In the study by Wang et al. [42], the muscle spindle firing rate at 50% motoneuron recruitment was identified as the most sensitive neural parameter investigated, especially when comparing individuals with mild and severe spasticity.

Such a spindle firing rate at 50% motoneuron recruitment represented a threshold for the minimum spindle firing rate [40,42,55] and was smaller in severe than moderate and mild spasticity volunteers [55]. It was lower in stroke patients with a more pronounced reflex contribution than passive components [52]. This rate significantly increased 4 weeks after BoNT-A application compared to the baseline values. Still, the spindle firing rate was reduced 12 weeks after treatment, indicating a transient effect of BoNT-A on spasticity [55]. The reduced hyperactive reflex was also attributed to an increased spindle firing rate at 50% motoneuron recruitment [55]. The cumulative distribution function of the alpha motoneuron pool showed a reduction with increasing spasticity [42] and decreased in volunteers with more pronounced reflex contributions [52]. However, the cumulative distribution function of the alpha motoneuron remained constant before and after BoNT-A intervention in the study by Wang, Gäverth, and Herman [55].

The damping coefficient was higher in the spasticity groups than in the healthy group [42,51,52,54], and it increased with the severity of spasticity [42,51,52]. In these studies, the damping coefficient corresponds to joint viscosity during passive movements [42,51,52,54].

In some cases, the elastic component is proportional to the angle or muscle length and, therefore, one has a linear stiffness coefficient [38,51]. When the elastic component depends exponentially on the angle or muscle length, one has a nonlinear stiffness coefficient [49,52,53]. Some studies incorporate both linear and nonlinear elastic components [42,55]. The linear stiffness coefficient of the joints was larger in severe and moderate than in mild spasticity or health conditions [42] but similar in healthy individuals and mild spasticity [42,54]. Similarly, the nonlinear stiffness coefficient of the joints increased with spasticity severity [42,51]. The nonlinear stiffness coefficient significantly increased after 12 weeks of BoNT-A application compared to baseline and 4 weeks post application values [55]. In the work of Shin et al. [52], the nonlinear stiffness coefficient increased in the subjects with a larger passive and reflex torque compared to the healthy group [52].

**The physiological plausibility of the models**

Table 4 shows the models' plausibility scores according to the investigated tasks. We provide supplementary material that presents the individual values obtained by the models according to the groups, tasks, muscles, and joints evaluated (S3 Table).

The model with the highest plausibility score was the one proposed by Wang et al. [42] which investigated the passive stretching of wrist flexors at controlled velocity using an external device at slow and fast velocities. In this model, proprioceptive afferents send information regarding joint angular position and joint angular velocity [42]. A compartmental musculotendinous unit incorporates muscle spindles and a motoneuron pool, allowing the identification of neural and non-neural parameters [42]. Two other models proposed for passive stretching at controlled velocity, using a framework similar to Wang et al. [42], also presented a nice plausibility index (with a total of 3 points each) [52,55].

A feedback model for investigating gait related to force and force derivative, proposed by Falisse et al. [43] also reached three points. Two types of musculoskeletal models were tested, but the best plausibility score was related to a model that personalized optimal fiber lengths and tendon slack lengths to the characteristics of the evaluated child [43]. This study revealed that altered muscle-tendon properties, rather than reduced neuromuscular control complexity and increased spasticity, were the primary cause of the crouch gait for one investigated child [43].

Considering the pendulum test, Kim, Eom and Hase's [49] model yielded the best performance. The criteria attended by most pendulum test models was the ability to identify the non-neural components of the increased resistance. Some models did not score [38,39,46,47] or had low global scores [40,44,49,50,53,54]. The main reason was the lack of quantitative information on the model's goodness of fit for experimental or literature data. Some papers described only quantitatively the goodness of the fit, reporting: "reasonably good fit" [38,46,54], "acceptable fit" [49] and "good fit" [53]. Other aspects linked to low scores were the lack of personalized anatomical geometries from medical images [38,41,44,47–51,53–55] and the incapacity to distinguish the origin of the different neural [38,39,41,43,45–51,53,54] and non-neural [38–40,43,45,47,48,50,53] mechanisms of joint hyper-resistance

## Discussion

This scoping review identified 18 primary studies that addressed neuromusculoskeletal modeling research on spasticity in various pathological conditions, published between 1997 and 2023. As far as we know, this is the first scoping review that summarizes the state of the art on neuromusculoskeletal modeling of spasticity. The discussion is organized according to the questions raised in this review, clinical implications, and future perspectives.

**What physiological phenomena are represented in the neuromusculoskeletal models?**

From a physiological point of view, this review did not identify any studies that fully characterize the structures involved in spasticity. None represent explicitly the involvement of supraspinal structures in spasticity (motor cortex, brainstem, cerebellum, and basal ganglia) or the ascending and descending sensory pathways, as demonstrated in Fig 3 and Table 1. Despite these limitations, there have been attempts to emulate supraspinal influences, adding a term representing supraspinal drive [45], a baseline torque [53], and neural weakness [44]. Notably, the studies did not establish a relationship between the included terms and specific supraspinal areas.

Mathematical formulations of spinal reflex mechanisms have played a central role in the investigation of spasticity [38–50,52,54] despite their relative simplicity compared with neuronal network models containing soma and dendrites and several motor neurons [75–77]. Most of these studies hypothesized that type Ia afferent fibers [38–44,46–50,52–55] are vital in generating spasticity. At the same time, some of them suggest that type II fibers might also be relevant [38,39,44,46,47,52]. In most models, the reflex loop feedback were the muscle fiber length and muscle stretching velocity, aligning with well-established physiology principles [59,60]. Some studies focus exclusively on the velocity dependence of the stretch reflex [41,48,49], as proposed by Lance's classic definition [8]. However, recent neuromusculoskeletal models suggest that muscle spindles could be sensitive to another type of sensory information, such as muscle force, muscle

force rate, and muscle fiber acceleration [43,50,53]. This perspective highlighted the need for comparative analysis among the available models, especially in cerebral palsy [43,50,53].

Falisse et al. [43,50] based their model on feedback from muscle force and its first derivative, following Blum et al. [78]. Blum et al. found from experiments on cats that the firing rates of Ia afferents were better predicted by muscle force than length, particularly at higher stretching velocities [78]. Falisse et al. [50] compared three neuromusculoskeletal models with distinct sensory feedback in children with cerebral palsy. They found that the model based on muscle force and its first derivative was better at predicting hamstring muscle activity during fast passive movements and gait compared to the models based on feedback from muscle fiber length and velocity (velocity-related model) and the model relied on feedback from muscle fiber length, velocity, and acceleration (acceleration-related model) [50]. Additionally, de Groote et al. [53] found that muscle force-based feedback, compared with velocity feedback in the stretch reflex, better reproduced the kinematics of the pendulum test under spasticity conditions.

**Table 4. Physiological plausibility score for the developed models according to the investigated tasks.**

| Task Type | Authors | Feedback Model | i | ii | iii | iv | v | NF | Physiological plausibility Score (Maximal Score Possible = 5) |
|---|---|---|---|---|---|---|---|---|---|
| PASSIVE STRETCHING | van der Krogt et al., 2016 [41] | Muscle stretching velocity | 0 | 0 | 0 | 0 | 4 | 4 | 1 |
| | Falisse et al., 2018 [50] | Muscle fiber length and muscle stretching velocity | 0 | 0 | 0 | 0 | 0 | 2 | 0 |
| | Falisse et al., 2018 [50] | Muscle force and muscle force rate | 1 | 2 | 0 | 0 | 0 | 2 | 1.5 |
| | Falisse et al., 2018 [50] | Muscle fiber length, muscle stretching velocity, and muscle stretching acceleration | 0 | 0 | 0 | 0 | 0 | 2 | 0 |
| | Shin et al., 2020 [52] | Muscle fiber length and muscle stretching velocity | 3 | 0 | 0 | 3 | 3 | 3 | 3 |
| | De Vlugt et al., 2011 [51] | Not specified | 8 | 0 | 0 | 0 | 8 | 8 | 2 |
| | Koo; Mak, 2006 [40] | Muscle fiber length and muscle stretching velocity | 0 | 0 | 0 | 1 | 0 | 1 | 1 |
| | Wang et al., 2017 [42] | Joint angular position and joint angular velocity | 3 | 3 | 0 | 3 | 3 | 3 | 4 |
| | Wang; Gäverth; Herman, 2018 [55] | Joint angular position and joint angular velocity | 6 | 0 | 0 | 6 | 6 | 6 | 3 |
| PENDULUM TEST | He; Norling; Wang, 1997 [46] | Muscle fiber length and muscle stretching velocity | 0 | 0 | 0 | 0 | 0 | 1 | 0 |
| | He, 1998 [38] | Muscle fiber length and muscle stretching velocity | 0 | 0 | 0 | 0 | 0 | 1 | 0 |
| | Fee; Foulds, 2004 [48] | Joint angular velocity | 0 | 0 | 0 | 0 | 4 | 4 | 1 |
| | Kim; Eom; Hase, 2011 [49] | Joint angular velocity | 0 | 1 | 0 | 0 | 1 | 1 | 2 |
| | Le Cavorzin et al., 2001 [54] | Joint angular position and joint angular velocity | 0 | 0 | 0 | 0 | 1 | 2 | 1 |
| | De Groote et al., 2018 [53] | Joint angular position and joint angular velocity | 0 | 0 | 0 | 0 | 1 | 1 | 1 |
| | De Groote et al., 2018 [53] | Joint torque and joint torque rate | 0 | 0 | 0 | 0 | 1 | 1 | 1 |
| | Feng; Mak, 1998 [47] | Muscle fiber length and muscle stretching velocity | 0 | 0 | 0 | 0 | 0 | 2 | 0 |

*(Continued)*

**Table 4.** (Continued)

| Task Type | Authors | Feedback Model | i | ii | iii | iv | v | NF | Physiological plausibility Score (Maximal Score Possible = 5) |
|---|---|---|---|---|---|---|---|---|---|
| GAIT | Falisse et al., 2018 [50] | Muscle fiber length and muscle stretching velocity | 1 | 1 | 0 | 0 | 0 | 2 | 1 |
| | Falisse et al., 2018 [50] | Muscle force and muscle force rate | 2 | 2 | 0 | 0 | 0 | 2 | 2 |
| | Falisse et al., 2018 [50] | Muscle fiber length, muscle stretching velocity, and muscle stretching acceleration | 0 | 0 | 0 | 0 | 0 | 2 | 0 |
| | Jansen et al., 2014 [39] | Muscle fiber length and muscle stretching velocity | 0 | 0 | 0 | 0 | 0 | 1 | 0 |
| | Falisse et al., 2020 [43] | Muscle force and muscle force rate | 1 | 1 | 1 | 0 | 0 | 1 | 3 |
| | Falisse et al., 2020 [43] | Muscle force and muscle force rate | 1 | 1 | 0 | 0 | 0 | 1 | 2 |
| | Bruel et al., 2022 [44] | Muscle fiber length and muscle stretching velocity | 0 | 0 | 0 | 1 | 1 | 1 | 2 |
| | Veerkamp et al., 2023 [45] | Muscle stretching velocity | 0 | 0 | 0 | 0 | 0 | 3 | 0 |
| | Veerkamp et al., 2023 [45] | Muscle force | 0 | 0 | 0 | 0 | 0 | 3 | 0 |

*Note.* Criteria: (i) good tracking of moments, (ii) good tracking of muscle activity, (iii) considerations about personalized geometries (obtained from medical images), (iv) distinguishes the neural origin of spasticity: spindle hypersensitivity (due to increased firing rate of muscle spindles) from motor neuron hyperexcitability and (v) distinguishes neural from non-neural components. NF = Normalization Factor.

In a recent study, Veerkamp et al. [45] contradicted the findings of Groote et al. [53] and Falisse et al. [50]. In gait simulations of children with cerebral palsy, the model with velocity feedback performed better than the one with force feedback in three identified gait patterns [45] in three identified gait patterns [45], although none of them scored in the criteria (i) and (ii) of the plausibility score.

Therefore, even among gait studies on cerebral palsy [43,45,50], disagreements exist regarding which model best reproduces spasticity characteristics. These discrepancies may arise from sample heterogeneity and differing modeling approaches. For instance, Falisse et al. [50] examined children with less severe spasticity than those studied by Veerkamp et al. [45]. Additionally, Falisse et al. [50] modeled muscle force feedback and its first derivative, whereas Veerkamp et al. [45] excluded the derivative due to its sensitivity to noise. These varying assumptions hinder consensus on the optimal spasticity model and highlight the need to harmonize study characteristics and theoretical assumptions, especially when aiming to identify the best way to model spasticity within the same pathology.

Furthermore, the variety in selecting different types of feedback related to the reflex circuit prompts us to reconsider how well we understand the role of muscle spindles concerning spasticity. Debates about the role of spindles in spasticity extend beyond neuromusculoskeletal modeling; they also occur in the neurophysiology field, with arguments that the traditional view of spindles is outdated [79,80]. Recognizing that muscle spindles are peripheral sensory organs innervated by gamma motor neurons, some authors suggest they primarily enhance sensorimotor performance rather than serve exclusively for proprioception [79,80]. Therefore, it's plausible that these discussions will permeate neuromusculoskeletal modeling. Applications in this area should consider these debates to deepen our understanding of spasticity, an equally complex phenomenon.

Taking into account the categories i) and ii) of the physiological plausibility score, the models related to muscle fiber length/joint angular position and muscle stretching velocity/joint angular velocity have the best scores [42,52,55]

compared to force-related models [43,50,53]. Studies addressing alternative types of feedback got low plausibility scores due to poor goodness of fit criteria. Therefore, further research is needed to determine the best ways to capture the characteristics of spasticity.

Considering the physiological plausibility of the models, those with more detailed and comprehensive descriptions achieved higher scores than simpler ones [42,52,55]. These higher scores were associated with models investigating passive tasks, most of which could reproduce experimental data and distinguish between neural and non-neural contributions to joint hyper-resistance [42,52,55]. In contrast, gait-related models limitedly reproduced experimental data and did not consider non-neural contributions to increased joint resistance [39,43,45,50]. Meanwhile, studies concerning the pendulum test often failed to quantitatively report measurements for criteria (i) and (ii) [38,46–48,53] but more frequently met criterion (v) [48,49,53,54].

As neuromusculoskeletal models can be used for clinical decision-making, all validation information should be fully reported. Despite the good scores of the models related to passive tasks, there are still drawbacks related to criterion (iii), which refers to individual parameter adjustments, met only by [43]. Additionally, musculoskeletal changes may occur secondary to the primary injury. Such occurrences have been little explored in neuromusculoskeletal models of spasticity, making it clear that acquiring in vivo measurements at the muscular level remains challenging.

### How do the models represent pathophysiology when considering neural and non-neural contributions?

In this review, we found studies related to several pathologies, including multiple sclerosis [38,46], pure spastic quadriplegia [47], stroke [39,42,49,51,52,54,55], and cerebral palsy [41,43,45,48,50,53] (Table 3), although the focus of the reviewed studies was cerebral palsy and stroke manifestations on the lower limb.

The identified studies suggest a common mechanism for spasticity, the hyperexcitable stretch reflex, as proposed in the classic Lance definition [8]. As discussed in the next topic, some musculoskeletal models' parameters differ according to the underlying pathology. Therefore, neuromusculoskeletal modeling might improve understanding of the underlying physiopathology mechanisms and identify disease-specific patterns.

Some studies have examined a range of diseases within a single investigation [44,45,52,54], but none have specifically explored the different underlying disease-specific mechanisms. Neuromusculoskeletal modeling can help to differentiate these mechanisms, exploring the various causes of spasticity by investigating each parameter for each disease. Furthermore, such models must be sufficiently detailed, considering all relevant parameters and variables to distinguish the different situations clearly.

In cases related to cerebral palsy, all models represented the stretch reflex mechanisms, but there was no consensus regarding the mechanical variable fed back in the reflex loop [45,50,53]. Therefore, the current models do not allow us to infer if and which mechanical variables are encoded by the spindles. In fact, as already mentioned, there are debates on this topic in the field of neurophysiology [79,80]. Therefore, it is essential that studies in this area are also conducted.

Some studies have also proposed mechanisms linked to spasticity at the spinal level, demonstrating that changes in motor neurons can occur [40,42,52,55]. These are supported by literature describing spinal-level changes, such as hyperexcitability of alpha motoneurons amplified by persistent calcium ($Ca^{2+}$) and sodium ($Na^+$) currents, leading to prolonged depolarizations [26]. The reviewed models do not consider these phenomena in detail but proposed lumped parameters to represent the net excitatory and inhibitory inputs to the motoneuron pool, such as MN pool threshold and gains. In the future, representing motoneurons more realistically could enhance understanding of spinal-level alterations.

Surprisingly, although not our initial focus, most studies considered secondary changes due to spasticity in tissues adjacent to the muscle [41,42,48,49,51–55] and within the muscle itself [43,44]. These findings highlight the importance of accounting for non-neural contributions to joint hyper-resistance beyond spasticity. The literature also describes tissue changes in people with spasticity [24,81,82]. We believe these investigations are justified by the need to distinguish

tissue-related contributions that increase joint resistance, especially since most studies involve individuals with chronic conditions — the participants had injuries for more than 10 months, up to 140 months post-injury. Identifying these different contributions can enable more targeted treatments tailored to each patient's specific needs.

**Which parameters have been able to quantify spasticity and to distinguish severity levels?**

As indicated in Table 2, the studies have proposed several parameters to quantify spasticity and other factors contributing to joint hyper-resistance. However, only a few of them were able to detect different levels of spasticity or distinguish between healthy and pathological groups. This might indicate that only few parameters are related to spasticity.

The pendulum test studies revealed that the thresholds associated with stretching velocity, position-dependent velocity, and muscle length were lower in people with spasticity than controls [47,54]. These results align with Tonic Stretch Reflex Threshold (TSRT) tests, which are also lower in spastic patients [83,84]. TSRT assigns the angle or length at which a muscle begins to activate in response to a stretch [85] is considered a biomarker of muscle activation and control impairment in individuals with neurological lesions. TSRT is often reduced in spastic muscles, leading to earlier activation of the muscle in response to passive stretching [83,85,86]. In [49], the threshold was linearly correlated with the EMG duration and the sum of the peaks, supporting the idea that spasticity results from the decrease in the stretch reflex threshold. Therefore, thresholds appear to be a valuable parameter for quantifying spasticity within the context of the neuromusculoskeletal model of the pendulum test.

Length and velocity feedback gains were higher in spasticity groups compared to healthy conditions, as found in cases of pure spastic quadriplegia [47], cerebral palsy [48], and in simulation studies related to gait [39,44]. Van der Krogt et al. [41] found that gains differed among volunteers with cerebral palsy during passive tasks and associated this result with children's severity levels. However, stroke-related studies also in passive tasks showed that feedback gains did not significantly differ across the levels of spasticity [42,49], and no changes were detected before and after antispastic intervention with BoNT-A [55]. Wang et al. calculated the sensitivity index of static spindle gain. They found it was not a significant parameter [42], contrasting with Koo and Mak [40], who identified the sensitivity index as significant. In the latter study, the etiology of spasticity was not described [40].

These different results of the contribution (or not) of the feedback gains to the altered patterns related to spasticity could be explained by spasticity etiologies. Some studies assume that the gains represent the level of gamma motoneuron activation [40,42,52,55], while others omit the origin of the feedback gains [39,44]. Based on EMG recordings, Forman et al. [87] suggest the underlying mechanisms of sustained involuntary muscle activity differ between cerebral palsy and stroke despite similar clinical presentations. In cerebral palsy, sustained involuntary muscle activity exhibits greater central or cortical contributions, indicated by higher gamma band coherence [87]. Conversely, in stroke, this activity shows more peripheral or afferent contributions, with position-dependent increases and higher alpha band coherence [87]. These findings seem to support the absence of feedback gain contributions to spasticity in stroke neuromusculoskeletal models [42,55], since most studies assume these feedback gains represent gamma motoneuron activation levels [40,42,52,55]. Similarly, if feedback gains were related to gamma motor neurons, increasing gains to elicit spasticity in cerebral palsy models would be justified, as a greater sustained activation with the coherence of gamma motor neurons was reported [87].

Two studies related to gait also reported that increased feedback gains were necessary to replicate altered movement patterns [39,44]. One was related to hemiparetic gait in stroke [39], while the other referred to any neurological impairment [44]. In stroke, feedback gains differed between gait [39] and passive tasks [42,55]. This suggests that the nature of the task influences the behavior of the feedback parameters. Thus, further studies on neuromusculoskeletal models of spasticity during gait and other active tasks are needed to enhance understanding of this issue. Considering the findings discussed above, future research needs to be more specific in identifying distinct behaviors for each etiology and task, perhaps making different model assumptions for each case.

Some parameters indicate that post-stroke spasticity is related to changes at the motoneuron pool level, even after intervention with BoNT-A. The MN pool threshold differentiates healthy conditions from varying levels of spasticity, being smaller in more severe cases [42]. Stroke patients exhibited increased MN pool threshold values four weeks after BoNT-A treatment, which returned to initial levels after 12 weeks [55]. Despite being a lumped parameter [40,42,52,55], this threshold can reveal differences in spasticity levels and post-intervention states [42,55]. Similarly, the MN pool gain was larger in groups with severe spasticity than in moderate and healthy conditions [42], but no changes were observed post-BoNT-A treatment [55]. Significant changes were noted in muscle spindle firing rates at 50% of motoneuron recruitment, which were lower in severe spasticity or individuals with greater reflex contributions [42]. Moreover, Wang, Gäverth, and Herman [55] specifically demonstrated that this parameter modifies after BoNT-A intervention, aligning with clinical studies.

BoNT-A acts in the neuromuscular junction, blocking the release of acetylcholine, causing muscle functional denervation and, thus, relaxation [88]. However, substantial evidence shows that BoNT-A also acts on the central nervous system [89,90]. Initially, the central effects were assumed indirect, resulting from modifying sensorimotor integration after peripheral application of the toxin [91]. However, recent studies in stroke patients [89,90] indicate that BoNT-A can be retrogradely transported along motor axons to the central nervous system, where it exerts direct effects, such as modification of spinal recurrent inhibition. Changes in muscle spindle firing rates at 50% motoneuron recruitment may be related to alterations in sensory feedback, as BoNT-A can modify sensory feedback by reducing Ia afferent fiber traffic [88,92]. This may affect the central nervous system, increasing muscle spindle firing rates at 50% motoneuron recruitment after BoNT-A intervention [55]. However, it is noteworthy that this parameter lacks a clear physiological explanation in the studies we identified [40,42,52].

Except for pendulum test studies, models that achieved the highest physiological plausibility scores [42,52,55] (Table 4) included parameters capable of expressing different levels of spasticity or distinguishing neural from non-neural contributions between groups. These studies demonstrated that parameters related to motoneurons behaved similarly across different stroke studies, indicating consistency in the results regarding neural alterations.

The proposed non-neural parameters showed similar behavior across most studies. Nonlinear stiffness was higher in groups with more severe spasticity than those with mild spasticity and healthy controls [42,51,52,55]. The damping coefficient was also elevated in groups with more severe spasticity compared to mild spasticity and healthy individuals [42,52,54]. However, there was no significant difference in this parameter between healthy individuals and those with mild spasticity [42,51]. Experimental elastography studies have also reported statistically significant differences in muscle stiffness measurements between affected and unaffected sides in people with spasticity [93–96] and between patients and healthy individuals [97,98]. Similarly, neuromusculoskeletal models that quantified non-neural parameters produced comparable results, highlighting their potential to aid in better defining treatments and distinguishing the different contributions to increased joint resistance.

Regarding changes after intervention, Wang, Gäverth, and Herman [55] reported that the nonlinear stiffness coefficient significantly increased 12 weeks after BoNT-A application compared to baseline and 4-week post-application values. This increase was accompanied by a significant reduction in the wrist range of motion at 12 weeks [55]. Studies on stroke patients' upper limbs following BoNT-A application showed improved clinical manifestations, including an increased joint range of motion and reduced stiffness [99,100]. However, these findings are limited to up to 4 weeks post-application, preventing the inference of long-term changes. A rat study found that passive stiffness in the limb injected with BoNT-A increased at 2,4, and 8 weeks [101], like Wang, Gäverth and Herman's findings [55]. These findings indicate that modeling may be a promising way to explore in more detail the mechanisms of action of interventions at different levels.

## Clinical implications

The neuromusculoskeletal models identified in this review effectively captured spasticity-related changes across various underlying pathologies. These models were sensitive in distinguishing parameters between individuals with and without

spasticity; some could even differentiate degrees of spasticity and identify non-neural contributions to increased joint resistance. These findings highlight the potential application of neuromusculoskeletal modeling strategies to support diagnostics and guide personalized treatments. As demonstrated in one study, these models can also evaluate the effects of interventions.

Implementing neuromusculoskeletal models in clinical practice requires multidisciplinary collaboration. Integrating engineers, scientists, clinicians, and therapists ensures these tools are technically robust, practical, and accessible for everyday clinical use.

## Future perspectives

Future studies should expand neuromusculoskeletal modeling of spasticity to include supraspinal mechanisms and more detailed spinal mechanisms, modeling motoneuron structures more realistically, as has been done in studies on degenerative diseases [30]. Although challenging, this could confirm the various mechanisms contributing to spasticity across different pathologies, as suggested by some studies in this review. Given the multifactorial and complex nature of spasticity, future studies must thoroughly describe the characteristics of the target population, and the performance measures used to validate the models, which were not consistently reported in the analyzed studies (Table 4). It is essential to verify if different etiologies might have distinct mechanisms. Therefore, analyzing these alterations in a grouped manner may lead to inaccurate generalizations, which should be avoided in future studies.

Among the studies reviewed, only one utilized magnetic resonance imaging (MRI) to customize muscle parameters [43]. Imaging techniques such as MRI and ultrasound can directly measure muscle volumes, force-tension parameters, and pennation angles, while elastography and myotonometry can quantify tissue stiffness [102–106]. These methods offer valuable comparisons with results from mathematical models. However, there is still a significant gap in customizing spasticity-related models, reflecting the limited number of research groups focused on this development, as noted by Fregly [106]. More detailed experimental data, including elastography and MRI measurements, could enable more accurate customization of muscle models.

Given the complexity of neuromusculoskeletal computational modeling and its application to studying spasticity, future models must provide detailed reports on data collection, validation methods, quantitative measures of model fit, and sensitivity indices of investigated parameters. Reporting this information will help identify which parameters and models are most sensitive, aiding in developing increasingly accurate models.

While extensive research focuses on changes during passive movements, studies on voluntary motor tasks are relatively scarce. This seems peculiar to spasticity, as noted in a review of technology-assisted spasticity assessment methods [32], who also identified more studies related to passive tasks. Probably this is so, due to the definition and clinical assessment of spasticity, which involves only passive movement. However, spasticity can occur during active movements in daily life [18,107], impairing the functionality of the individuals. Thus, the clinical community would profit from a better understanding of spasticity during active tasks and neuromusculoskeletal modeling should better address this issue. Another issue not explored yet is the difference between affected and unaffected limbs in cases of hemiplegic and hemiparetic individuals.

## Study limitations

Our use of specific keywords may have limited our search strategy for identifying articles on neuromusculoskeletal modeling and spasticity. To minimize the risk of missing relevant studies, we manually searched the reference lists of selected studies and reviews published during this study's development. This approach allowed us to include additional studies that met our inclusion criteria.

We did not consider age differences, aiming instead to identify neuromusculoskeletal models that represent spasticity arising from diverse etiologies. However, age may imply differences in the muscular and skeletal systems. Future reviews should consider these aspects. The same applies to the identification of sensitive parameters to quantify neural aspects



of spasticity. Our focus was on identification rather than assessing the accuracy of the proposed parameters. We suggest that future studies investigate these measures to determine which are truly useful for assessing components of spasticity and other contributions of joint hyper-resistance.

## Supporting information

**S1 Table. Preferred Reporting Items for Systematic reviews and Meta-Analyses extension for Scoping Reviews (PRISMA-ScR) Checklist.**
(DOCX)

**S2 Table. Search strategy used for each of the databases.**
(DOCX)

**S3 Table. Physiological plausibility score for the developed models according to the investigated tasks, target muscles and/or joints, and evaluated groups.**
(DOCX)

**S4 Table. Characteristics of the included studies concerning funding and competing interests.**
(DOCX)

## Author contributions

**Conceptualization:** Verônica Andrade da Silva, Elisangela Ferretti Manffra.

**Data curation:** Verônica Andrade da Silva, Rafael Lucio da Silva, Joseana Wendling Withers, Kátia Janine Veiga Massenz, Maria Isabel Veras Orselli, Elisangela Ferretti Manffra.

**Formal analysis:** Verônica Andrade da Silva, Rafael Lucio da Silva, Joseana Wendling Withers, Kátia Janine Veiga Massenz, Maria Isabel Veras Orselli, Luciano Luporini Menegaldo, Elisangela Ferretti Manffra.

**Funding acquisition:** Elisangela Ferretti Manffra.

**Investigation:** Verônica Andrade da Silva, Rafael Lucio da Silva, Joseana Wendling Withers, Kátia Janine Veiga Massenz, Maria Isabel Veras Orselli.

**Methodology:** Verônica Andrade da Silva.

**Project administration:** Elisangela Ferretti Manffra.

**Supervision:** Maria Isabel Veras Orselli, Luciano Luporini Menegaldo, Elisangela Ferretti Manffra.

**Validation:** Maria Isabel Veras Orselli, Luciano Luporini Menegaldo, Elisangela Ferretti Manffra.

**Writing – original draft:** Verônica Andrade da Silva.

**Writing – review & editing:** Verônica Andrade da Silva, Maria Isabel Veras Orselli, Luciano Luporini Menegaldo, Elisangela Ferretti Manffra.

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
