## [Decision Letter · Decision Letter 0]

14 Feb 2025

Neuromusculoskeletal modeling of spasticity: a scoping review

PONE-D-24-45090

Dear Dr. DA SILVA,

We’re pleased to inform you that your manuscript has been judged scientifically suitable for publication and will be formally accepted for publication once it meets all outstanding technical requirements.

Kind regards,

Yih-Kuen Jan, PhD

Academic Editor

PLOS ONE

Additional Editor Comments (optional):

Reviewers' comments:

Reviewer's Responses to Questions

**Comments to the Author**

1. Is the manuscript technically sound, and do the data support the conclusions?

Reviewer #1: Yes

Reviewer #2: Yes

2. Has the statistical analysis been performed appropriately and rigorously? 

Reviewer #1: Yes

Reviewer #2: Yes

3. Have the authors made all data underlying the findings in their manuscript fully available?

Reviewer #1: Yes

Reviewer #2: Yes

4. Is the manuscript presented in an intelligible fashion and written in standard English?

Reviewer #1: Yes

Reviewer #2: Yes

5. Review Comments to the Author

Reviewer #1: This is very well executed and valuable scoping review. The approach is well described and appropriate. The topic coverage is excellent. I have a few minor comments to consider:

1) at line 222, the statement "ability to distinguish the neural origin of spasticity ..." is a bit confusing. Do you mean that the distinction between spindle and MN origins is distinguished by the model? I am not sure how "ability" is judged in this context.

2) similar comment regarding line 225. How is "ability" judges in the statement "the ability ... to identify neural and non-neural components."

Reviewer #2: Because the article's methodological rigor demonstrates adherence to PRISMA-ScR, guaranteeing reproducibility and transparency, it is appropriate for publication. Additionally, because it advocates for etiology-specific approaches and identifies gaps in current models (such as the neglect of supraspinal mechanisms and active tasks), the article may have clinical relevance. Clear analysis parameters (such as stiffness and viscosity) that could improve biomechanical models are also displayed by the data synthesis.

6. PLOS authors have the option to publish the peer review history of their article (what does this mean? ). If published, this will include your full peer review and any attached files.

**Do you want your identity to be public for this peer review?** For information about this choice, including consent withdrawal, please see our Privacy Policy .

Reviewer #1: No

Reviewer #2: **Yes: ** Sanusi Mohammad Bello

---

## [Editor Report · Acceptance letter]

PONE-D-24-45090

PLOS ONE

Dear Dr. da Silva,

I'm pleased to inform you that your manuscript has been deemed suitable for publication in PLOS ONE. Congratulations! Your manuscript is now being handed over to our production team.

Kind regards,

on behalf of

Dr. Yih-Kuen Jan

Academic Editor

PLOS ONE